# Effective Removal of Biogenic Substances Using Natural Treatment Systems for Wastewater for Safer Water Reuse

**Wojciech Halicki** [1,2,*] and **Michał Halicki** [3]

1 Institute of Applied Ecology, Skórzyn 44a, 66-614 Maszewo, Poland
2 Centre for East European Studies, Faculty of Oriental Studies University of Warsaw, ul. Krakowskie Przedmieście 26/28, 00-927 Warszawa, Poland
3 Faculty of Earth Sciences and Environmental Management, University of Wrocław, pl. Uniwersytecki 1, 50-137 Wrocław, Poland
* Correspondence: w.halicki.ies@gmail.com

**Abstract:** Natural Treatment Systems for Wastewater (NTSW) show great potential for economic, socially acceptable and environmentally friendly wastewater treatment, along with the renewal of water for its safe reuse. This article presents the reduction in nitrogen and phosphorus compounds in domestic wastewater, which was achieved in a 2.5-year operation of the newly developed NTSW. The presented installation was developed by the Institute of Applied Ecology in Skórzyn (Poland) and implemented as a pilot plant serving the institute building with three permanent residents and up to five employees. The installation consisted of two parts, responsible for: wastewater treatment (septic tank and compost beds) and water renewal (denitrification beds, phosphorus beds and activated carbon beds). The mean values of nitrogen and phosphorus compound concentrations obtained in the renewed water for the entire research period were: 0.8, 49.4, 12 and 3.1 mg/L for ammonium nitrogen ($NH_4$), nitrates ($NO_3$), total nitrogen and phosphates ($PO_4$), respectively. Thus, average reductions of 99.6%, 90.9% and 94.4% were obtained for $NH_4$, total nitrogen and $PO_4$, respectively. Treatment of domestic sewage to such a level, similar to drinking water, enables versatile, safe water reuse, which in the situation of increasingly limited water resources will constitute increasing ecological and economic value.

**Keywords:** natural treatment systems for wastewater; wastewater treatment; water renewal; biogenic substances

## 1. Introduction

The global climatic conditions that have been changing intensively in recent decades will reduce the per capita water availability in European countries by more than ten percent between 1990 and 2050 [1]. Due to this fact alone, it is necessary to make numerous changes in water and wastewater management in order to guarantee access to water in the right quantity and quality for present and future generations. Access to good-quality water is necessary not only to meet municipal needs but also for the normal functioning of the environment, both globally and locally. For this reason, the words of René Dubos become more and more important: think globally, act locally [2]. Such an approach should guide the creation of a local sustainable water and wastewater management, which should include such elements as: (1) using, where possible, local water intakes for drinking water supply, and (2) wastewater treatment with water renewal and its full reuse in order to close local water circuits.

An approach promoting decentralized and sustainable water management has already been identified under Agenda 21 [3]. However, especially, the task of local wastewater treatment with water restoration and full reuse is difficult to achieve. Difficulties may arise from numerous reasons. First of all, construction and operation of local small sewage treatment plants is associated with high costs. Furthermore, installations for the renewal of

water and its reuse generate additional costs and increase energy consumption. This fact may significantly inhibit the implementation of local wastewater purification and water renewal, although such solutions are still cheaper than central wastewater systems with sewerage [4]. Other types of difficulties are technological challenges. Although there are currently many technologies on the market that guarantee effective wastewater treatment, there are only a few water renewal technologies that guarantee water of very good quality, the reuse of which is safe for people and the environment. There is also lack of experience with the multilateral reuse of reclaimed water. So far, water is reused mainly in agriculture and in watering urban greenery, e.g., [5]. Moreover, in the EU, official recommendations refer only to agricultural wastewater reuse [6]. This fact severely limits the introduction of the widespread use of renewed water.

The solution to these difficulties may be the use of various types of Natural Treatment Systems for Wastewater (NTSW). Such systems can be much cheaper to build, and especially to operate, thanks to, among other things, low energy demand. As numerous studies in the USA have shown, natural systems are more advantageous in these aspects compared with conventional mechanical–biological treatment plants [7]. Moreover, such systems favor the decentralization of water and sewage management, which is usually cheaper than central management [8]. Properly designed, built and operated NTSW can simultaneously treat sewage, renew water and, additionally, it is itself an ecological use of water by creating new habitats [9]. NTSW have been shown to be suitable for wastewater treatment in industry, agriculture and for individual households, e.g., [10–15]. Conventional water renewal plants, on the other hand, require additional purification installations, which not only increases investment and operating costs but also energy demand. In such situations, from an environmental point of view, the benefits of reusing water may be fewer than the negative environmental effects of the increased energy consumption.

The main risk related to the reuse of water is its low quality and, above all, the content of harmful substances such as heavy metals, biogenic substances and others [16]. For this reason, not every effluent from a treatment plant can be a source of safely reusable water. In order to minimize the risk to people and the environment, efforts should be made to ensure that the water intended for reuse is of the best possible quality. Ideally, it should meet the cleanest surface water criteria or even drinking water requirements. Such criteria cannot be met by conventional treatment plants without additional equipment. However, as indicated in numerous publications, NTSW can provide renewed water that meets such criteria [12,17]. Therefore, such treatment plants can be employed in places where it is planned to fully reuse the renewed water in a way that is safe for people and the environment [18].

The reclaimed water mainly comes from large central treatment plants and is reused mainly in agriculture in arid areas, yet there are many other local destinations for water reuse which can occur all year round and in different climatic zones, such as [19]:

- Impoundment reuse—the use of reclaimed water in an impoundment;
- Environmental reuse—the use of reclaimed water to create, enhance, sustain, or augment water bodies, including wetlands, aquatic habitats, or stream flow;
- Groundwater recharge–nonpotable reuse—the use of reclaimed water to recharge aquifers that are not used as a potable water source;
- Indirect potable reuse—augmentation of drinking water sources (surface or groundwater) with reclaimed water, followed by an environmental buffer that precedes normal drinking water treatment.

Each of the above-mentioned types of water reuse has direct or indirect effects on groundwater and surface water. Taking into account the present poor condition of groundwater and surface waters in Europe (e.g., in the EU as much as 24% of groundwater and 22% of surface waters are of poor quality [20,21]), it is necessary to ensure that the renewed water is of the best possible quality. Particularly important for the environment is the pollution of waters with nitrogen and phosphorus compounds [22–26]; therefore, the elimination of these substances from renewed water is an important challenge.

In this study, we describe the performance of an NTSW pilot plant over 2.5 years of continuous operation at the building of the Institute of Applied Ecology (Instytut Ekologii Stosowanej-IES) in Skórzyn, West Poland. The novelty of this plant resides in the use of compost beds, as well as in the separation of individual sewage treatment processes in separate beds, which include organic matter removal, denitrification and phosphorus removal. This plant was designed to purify domestic sewage and renew water to such an extent that the outflow meets drinking water quality criteria in terms of nitrogen compounds (ammonium nitrogen, nitrates and nitrites). In addition, phosphorus, the content of which is not regulated in drinking water, should be as low as possible. Only water of such quality enables complete, versatile water reuse that is safe for both people and the environment.

## 2. Materials and Methods

### 2.1. Description of the NTSW Pilot Plant

The research was carried out on a pilot plant consisting of two parts, responsible for wastewater treatment and water renewal, respectively. A detailed description of the treatment plant can be found in Halicki and Halicki [12]. The following description focuses on the elimination of biogenic substances. The pilot plant was built on the premises of the IES (http://ies.zgora.pl/en/home/, accessed on 16 October 2022) and is designed to treat sewage from the IES building with 3 residents and up to 5 office workers. The Institute is located in West Poland (52°7′15.808″ N, 15°2′21.102″ E). The setting of the plant can be found on the IES website. The pilot plant consists of a septic tank, compost beds, water renewal beds (i.e., denitrification, phosphorus elimination and activated carbon beds) and a buffer reservoir. Each beds facility is divided into three separate beds, working under a different daily hydraulic load, namely 70, 100 and 130 L/m² for the A, B and C beds, respectively (Figure 1). Pictures of the pilot plant are presented in Figure 2. The beds are built to allow the gravitational flow of water between the individual beds. All water renewal beds are covered with swamp vegetation.

- The septic tank with a capacity of 3 m³ receives sewage from the IES building in an average daily amount of 300 L, which results in a retention time of about 8 days. Mechanical and partly biological wastewater treatment takes place in the septic tank. After two years of operation, the septic tank is emptied of sewage sludge.
- The compost beds with a total area of 3 m² are placed in a greenhouse and divided into 3 separate parts (A, B and C) with an area of 1 m² each. These beds are 1 m high and consist of the following layers: on top, an 80 cm compost layer (80% wood chips and 20% peat soil), and below, a 10 cm layer of sand (granulation from 0.2 to 2 mm) and a 10 cm layer of gravel (granulation from 4 to 16 mm). Additionally, 10 kg of fertilizer lime containing 60% calcium carbonate was added to the compost beds. In terms of nutrient removal, the main task of the compost beds is to perform maximum ammonium nitrification, simultaneous denitrification of nitrate nitrogen and partial (biological and chemical) phosphorus elimination. In addition, the beds provide a very effective reduction in organic matter.
- The denitrification beds, with a total area of 3 m², are also divided into 3 equal parts (A, B and C). Each of the beds is supplied with sewage treated in the corresponding compost bed. Denitrification beds are 1 m high and consist of a 40 cm layer of swamp sediments, a 30 cm layer of sand (granulation from 0.2 to 2 mm) and a 10 cm layer of gravel (granulation from 4 to 16 mm). These beds are responsible for the further denitrification of nitrates flowing out of the compost beds.
- The phosphorus beds, like the previous ones, have an area of 3 m², are divided into 3 equal parts (A, B and C) and are supplied with the outflow from the corresponding denitrification beds. The beds are filled from the top with a 50 cm layer of sand (granulation from 0.2 to 2 mm, with the addition of 25 kg of construction lime) and a 10 cm layer of gravel (granulation from 4 to 16 mm). The main task of the phosphorus beds is to further reduce phosphorus compounds.

- The activated carbon beds, also with an area of 3 m$^2$ and divided into 3 equal parts (A, B and C), are supplied with the outflow from the corresponding phosphorus beds. The beds are filled from the top with a 50 cm layer of sand (granulation from 0.2 to 2 mm, with the addition of 25 kg of activated carbon) and a 10 cm layer of gravel (granulation from 4 to 16 mm). The main task of the activated carbon beds is to further reduce organic compounds.
- The last element of the pilot plant is the retention reservoir. It is an underground tank sealed with foil, filled with fine gravel (granulation from 2 to 4 mm) and covered with a layer of soil. The capacity of the tank is 5 m$^3$.

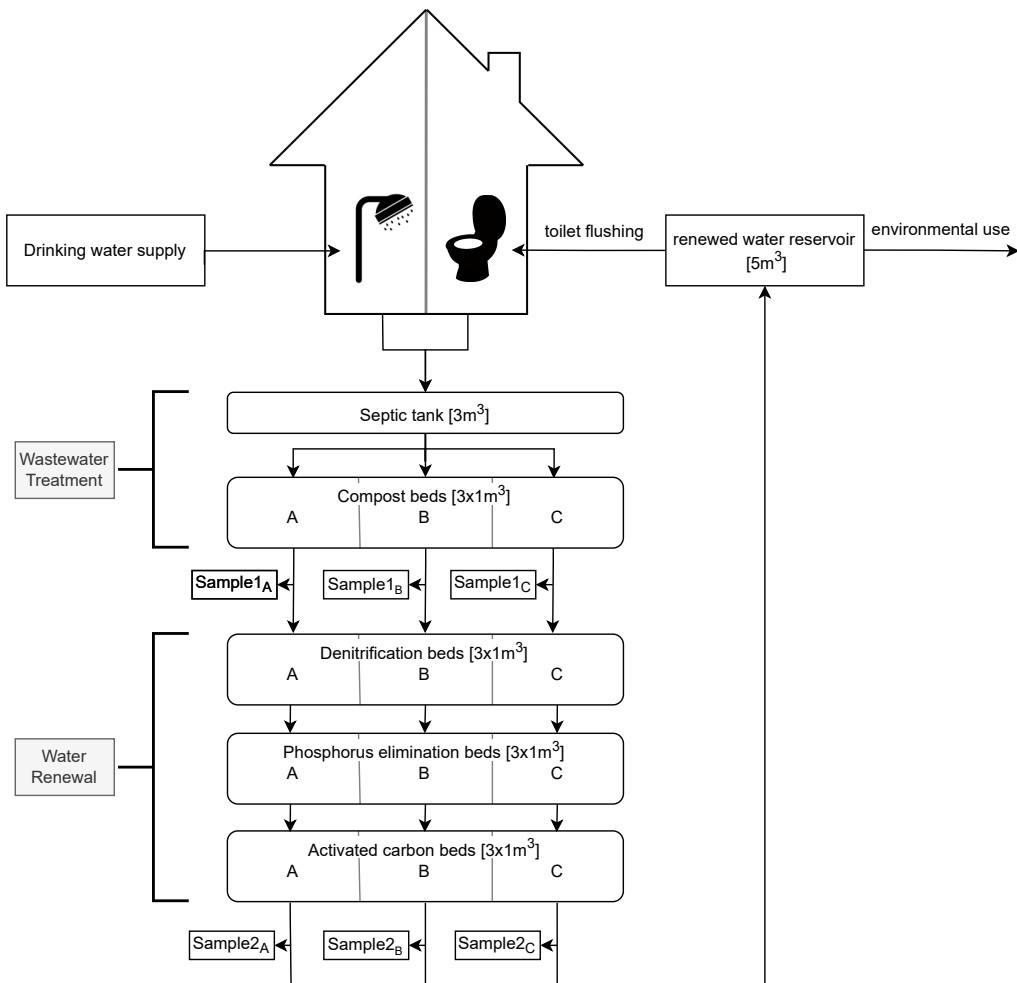

**Figure 1.** Scheme of the NTSW pilot plant (adapted from Halicki and Halicki [12]). The three parallel wastewater treatment and water renewal processes are conducted in different daily hydraulic loads, namely 70, 100 and 130 L/m$^2$ for the A, B and C letters, respectively.

## 2.2. Sampling and Quality Assessment

Raw sewage sampling took place once a month, while the sampling of treated sewage in the outflow of compost beds (Sample 1) and sampling of renewed water in the outflow of the activated carbon beds (Sample 2) took place every two weeks. The time span of this study is 2.5 years, ranging from August 2019 to May 2022. The analyses of nitrogen and phosphorus compounds were tested according to the Standards Methods for the Examination of Water and Wastewater [27], i.e., methods designated as 4500-NH$_3$ E, 4500-NO$_3$ C, 4500-NO$_2$ B and 4500-P C. For the above methods and to determine total nitrogen (TN) and total phosphorus (TP), the photolab 7100 VIS spectrophotometer was used. The measuring range of the methods employed are given in Table 1.

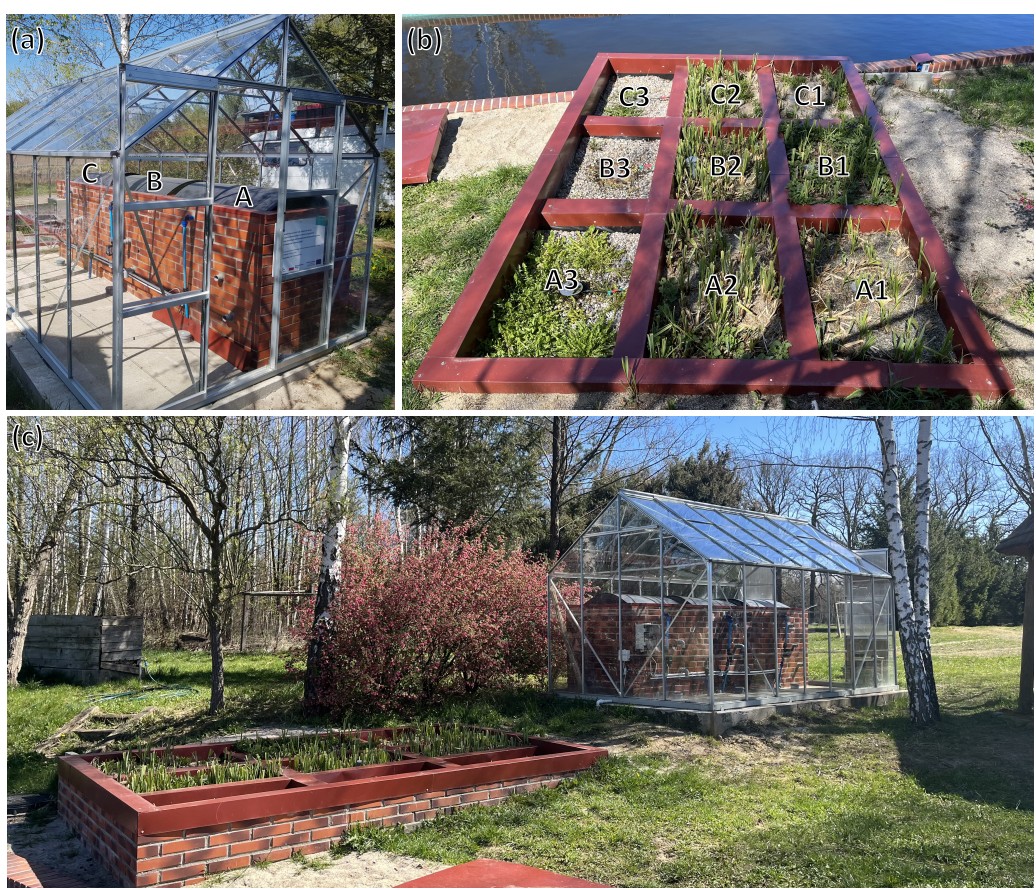

**Figure 2.** The NTSW pilot plant (adapted from Halicki and Halicki [12]). The three parallel wastewater treatment and water renewal processes are conducted in different daily hydraulic loads, namely 70, 100 and 130 L/m$^2$ for the A, B and C letters, respectively. (**a**) Compost beds. (**b**) Water renewal beds: denitrification beds (1), phosphorus beds (2) and activated carbon beds (3). (**c**) Setting of the NTSW pilot plant.

**Table 1.** Measuring range of the water quality assessment methods used in this study.

| Value | Range (mg/L) |
|---|---|
| $NO_2$ | 0.007–3.28 |
| $NO_3$ | 0.4–110.7 |
| $NH_4$ | 0.01–3 |
| $PO_4$ | 0.007–15.3 |
| Total Nitrogen | 0.5–15 |
| Total Phosphorus | 0.2–15.3 |

*2.3. Wastewater Treatment and Water Renewal*

The process of wastewater treatment takes place during its flow through the septic tank and compost beds, while the process of water renewal takes place in the denitrification, phosphorus and activated carbon beds (Figure 1). Removal of phosphorus and nitrogen compounds begins in the septic tank and takes place both at the stage of wastewater treatment and water renewal. However, the efficiency of these processes in the septic tank is low due to anaerobic conditions. In the septic tank, mainly suspended solids and organic substances are eliminated, mostly by their sedimentation and deposition at the bottom of the tank. Along with these compounds, part of the nitrogen and phosphorus is also deposited in the sediment. The septic tank ensures an average Hydraulic Residence Time (HRT) of 8 days. Sewage flows to the tank irregularly, both on a daily and weekly scale. The composition of the inflowing wastewater is also irregular. On the other hand, the

outflow from the septic tank is regular and takes place every hour. For this purpose, a pump controlled by a relay is used, which turns it on every hour for a specific time of operation. The outflow is directed to three compost beds (A, B and C), which work under different daily hydraulic loads of 70, 100 and 130 L/m$^2$, respectively.

The main process of wastewater treatment, including the elimination of nitrogen and phosphorus compounds, takes place in these beds. Compost is designed to create favorable conditions for the development of numerous and species rich soil and compost fauna and flora [28]. The primary function of compost beds is the elimination of organic matter. The presence of heterotrophic bacteria, actinomycetes, fungi, protozoa nematodes, vases, earthworms and others in compost beds contributes to the removal efficiency [29]. On the other hand, effective removal of organic compounds in compost beds provides the basis for the effective process of nitrogen removal. This process begins with the oxidation of ammonium nitrogen (i.e., nitrification) by autotrophic bacteria. Since these bacteria are absolute aerobes, they must be provided with an adequate amount of oxygen, which is consumed in the first place by heterotrophic bacteria that oxidize organic substances. In practice, this means that organic matter must first be removed from the wastewater, so that the heterotrophic bacteria do not consume oxygen. Then, autotrophic bacteria of the genera *Nitrosomonas* and *Nitrobacter* can develop, oxidizing the ammonium nitrogen contained in the wastewater to nitrites and nitrates. The sewage flowing through the compost beds is first depleted of organic matter and then, by means of nitrification, ammonium nitrogen is oxidized. However, the process of nitrogen elimination in compost beds does not end there. The compost used in the beds is made of particles/aggregates of various sizes. There are, however, various conditions in them, from aerobic through anoxic (facultative) to anaerobic. It is the anoxic conditions that favor the process of nitrates denitrification, while the organic compounds necessary for this process for facultative heterotrophic bacteria are available in the compost. Thus, in compost beds, nitrogen undergoes simultaneous nitrification and partial denitrification.

Phosphorus is eliminated in the compost beds in a biological and chemical way. Firstly, phosphorus contained in the treated wastewater is built into newly developing organisms which, after dying out and partial mineralization, increase the amount of organic matter containing phosphorus compounds. Furthermore, by adding fertilizer containing calcium carbonate to the compost beds, a process of chemical binding of some phosphates with calcium carbonate takes place, which results in the formation of sparingly soluble calcium phosphate.

The next step of nitrogen and phosphorus removal takes place during the water renewal process. In the first step, treated water flows through three denitrification beds, each connected to a corresponding compost bed (see Figure 1). In these beds, conditions are created that favor the development of facultative heterotrophic bacteria, which are responsible for removing nitrates and nitrites from the water in the process of denitrification. In the second stage, the renewed water flows through three phosphorus beds (A, B and C), which are fed by the outflow from the corresponding denitrification beds (see Figure 1). The beds provide conditions for the chemical bonding of phosphates with calcium carbonate compounds and partially with magnesium carbonate. The last stage of water renewal occurs in the activated carbon beds (A, B and C). Although the main task of these beds is to remove the remaining organic matter, they also contribute to a further reduction in ammonium nitrogen and phosphates that are being absorbed on the surface of the activated carbon. The renewed water is stored in a reservoir, which is only to perform a retention function. As the research presented in the next chapter shows, however, the processes of transformation of biogenic compounds take place in the reservoir to a small extent. Moreover, the renewed water from the three treatment paths (A, B and C) is of different quality. This water is mixed in the buffer tank and its composition is averaged.

## 3. Results

*3.1. Removal of Nitrogen and Phosphorus during the Wastewater Treatment Process*

### 3.1.1. Septic Tank

The following daily pollutant loads were assumed to calculate the reduction in nitrogen and phosphorus in the septic tank, and then in compost beds and water renewal beds: TN of 11 g and TP of 2 g per inhabitant [30]. On the other hand, the volume of sewage flowing into the septic tank was taken based on the measurement of the outflow from the septic tank, which was within the standard water consumption in Poland, ranging from 80 to 100 L/m per inhabitant and about 15 L per office worker [31]. During the week, the septic tank received sewage from three people living in the IES building and up to five office workers, which was about 340 L per day, while on the weekend the daily amount of inflowing sewage was on average 200 L. The average daily outflow from the septic tank was about 300 L. Based on the above data, it was assumed that the septic tank received pollutant loads corresponding to four equivalent inhabitants, which constituted daily loads of 44 g and 8 g for TN and TP, respectively. Thus, the average concentration of raw sewage was 147 g $N/m^3$ and 26 g $P/m^3$. Table 2 shows the average, minimum and maximum concentrations of nitrogen and phosphorus compounds in the outflow from the septic tank. Comparing the adopted mean values of concentrations in raw sewage before the septic tank and the concentrations in the outflow from the septic tank, we obtain a probable reduction of 11% for TN and 11.5% for TP, respectively. Converting the value of $NH_4$ in Table 2 to the value of $NH_4$-N, we obtain the concentration of ammonium nitrogen in the form of pure nitrogen at the level of 136 g, which is 6 g higher than the concentration of TN. This difference may be due to the lower accuracy of the TN determination method, which in fact should be at least slightly greater than the ammonium nitrogen expressed as $NH_4$-N. On the basis of this comparison, it can be assumed that in the outflow from the septic tank, practically all nitrogen is present in the form of ammonium nitrogen, while the remaining forms of nitrogen, such as organic nitrogen, nitrates and nitrites, constitute a minimal, insignificant amount. Converting the concentration of phosphates given in Table 2 as $PO_4$ to the value of $PO_4$-P, we obtain a concentration expressed in the pure component at the level of 19 mg/L. Comparing this value to the average concentration of TP from Table 2, it can be seen that in the case of phosphorus compounds, as much as 83% are present in the form of dissolved phosphates in sewage, and 17% are in a bound form in the organic matter contained in the sewage.

**Table 2.** Concentration of biogenic substances of raw sewage in the outflow of the septic tank.

| Value (mg/L) | Min | Max | Mean |
| --- | --- | --- | --- |
| $NH_4$ | 85.7 | 413.0 | 176.5 |
| $NO_3$ | 0.0 | 3.2 | 0.1 |
| $NO_2$ | 0.0 | 0.6 | 0.1 |
| Total Nitrogen | 90.0 | 168.0 | 130.7 |
| $PO_4$ | 9.2 | 81.7 | 55.3 |
| Total Phosphorus | 11.8 | 31.6 | 23.1 |

### 3.1.2. The Compost Beds

Table 3 shows the minimum, maximum and average concentrations of nitrogen and phosphorus compounds in the outflows from three compost beds and water renewal beds for the entire measurement period. On the other hand, Figure 3 shows the average monthly values of nitrogen and phosphorus concentrations in: (1) raw sewage discharged from the septic tank, (2) treated sewage discharged from three compost beds and (3) in water discharged from water renewal beds. As shown in Table 3, for the entire measurement period, the average concentration of ammonium nitrogen in the outflow from the three compost beds was 4.7 mg $NH_4$/L. When comparing this value to the mean ammonium nitrogen concentration in the septic tank outflow, the compost beds provided an average reduction of 97.4%. This reduction was based on the oxidation of ammonium nitrogen to

nitrates and nitrites in beds (ammonium nitrification). As a result, the nitrate concentration increased from the value of 0.1 mg $NO_3$/L in raw sewage to the average value after compost beds of 132 mg $NO_3$/L. The compost bed A showed the highest efficiency of the nitrification process, which operates at the lowest daily hydraulic load value of 70 L. In the outflow from this bed, the average $NH_4$ concentration was 2.7 mg/L, which gives a 98.5% removal efficiency. The other two beds, B and C, despite different hydraulic loads, achieved very similar average values of ammonium nitrogen concentration. Despite this fact, the fluctuations in the concentrations of ammonium nitrogen in the outflow from these beds varied significantly over the entire measurement period (Figure 3a). The greatest fluctuations occurred in the outflow of the bed B (daily hydraulic load of 100 L). Compost bed A worked most steadily, where the monthly average in the entire measurement period exceeded the value of 5 mg $NH_4$/L only once. The outflow from this bed was also characterized by a high stability of $NH_4$ concentration, independent of the season and the value of the concentration in the inflow to the bed (ranging from 100 to 400 mg/L). The nitrification of ammonium nitrogen resulted in the conversion of this compound into nitrates and nitrites.

As shown in Table 3, the mean values of nitrates in the outflow from the three beds differed only slightly. Moreover, in this case, the lowest concentration was achieved by compost bed A working with the lowest load, and the highest value for bed B (medium load). These slight differences in concentrations, despite the different values of the hydraulic load, occurred over the entire measurement period (Figure 3b). It is worth noting that from May 2021 there was a significant increase in nitrate concentrations, which lasted until February 2022, after which values in all beds began to decline. The marked increase in these concentrations partly coincided with the increase in the concentration of ammonium nitrogen in the inflow to the beds (Figure 3a). This probably resulted in an increasing concentration of nitrates in the outflow of beds. The concentration of nitrite in the outflow from all three beds was similar (about 1 mg $NO_2$/L); also, the monthly averages were stable throughout the entire measurement period (Figure 3c). This fact proves that neither the changing seasons nor the values of ammonium nitrogen in the inflow had a significant influence on the concentration of nitrites. Despite the very effective and stable nitrification of ammonium nitrogen in the compost beds, the process of denitrification took place simultaneously in them, as evidenced by the obtained values of TN in the outflow. The mean value of TN in the runoff from all compost beds was 42 mg/L, which, taking into account the mean value of the runoff from the septic tank (over 130 mg/L), indicates an average reduction in TN of 68%. All beds showed similar denitrification efficiency (Table 3). From August 2020, a significant increase in TN concentration began, which lasted until April 2021, after which the concentration stabilized at about 80 mg/L and did not change until the end of the measurement period, despite the fact that the concentration of ammonium nitrogen in the outflow from the septic tank decreased significantly (Figure 3d). This fact may indicate a decrease in denitrification potential that occurred in the second half of the measurement period.

While the removal of nitrogen compounds in the compost beds was only biological, the removal of phosphorus compounds was biological and chemical. The average value of phosphates in the treated sewage was 25.8 mg $PO_4$/L, which compared with the outflow from the septic tank at 55.2 mg $PO_4$/L gives an elimination in compost beds of 54%. In contrast, the elimination of TP in compost beds was 45%. It is worth noting that the hydraulic load had no effect on the elimination efficiency, which was similar and stable in all compost beds (Figure 3f). Only in the last months of the research period was there an increase in the concentration of TP in the outflow from compost beds.

### 3.2. Removal of Nitrogen and Phosphorus in the Process of Water Renewal

The first process taking place in denitrification beds is the removal of nitrates. As shown in Table 3, the average nitrate concentration after all beds is 49.4 mg/L. Compared with the average $NO_3$ concentration in the treated wastewater of 132 mg/L, the beds

ensured a reduction in nitrates of 63%. Interestingly, all three processes (A, B and C) provided a similar reduction efficiency, ranging from 61 to 63%. It is also worth emphasizing the stable concentration of nitrates in the outflow from all three beds over the entire measurement period (Figure 3a). As can be seen from this Figure, the increase in nitrate concentration in the runoff from compost beds that occurred since May 2021 did not affect the amount of $NO_3$ concentration in the runoff from water renewal beds. Equally stable values were ensured by the water renewal process with regard to the concentration of ammonium nitrogen and nitrites. In both cases, the concentration of ammonium nitrogen and nitrites remained below 1 mg/L throughout the entire measurement period. In the process of water renewal, in which the water flows through the denitrification, phosphorus and activated carbon beds, as much as an almost 85% reduction in ammonium nitrogen occurred. In contrast, $NO_2$ reduction was almost imperceptible due to the fact that the runoff from the compost beds was already very low (<1 mg/L). The reduction in TN during the process of water renewal in the beds was on average 72% and was at a similar level in each of the three processes (A, B and C). As it can be seen in Figure 3d, it was as stable throughout the entire measurement period as in the case of ammonium nitrogen.

The removal of phosphorus compounds took place mainly in phosphorus beds. During the water renewal process, the beds achieved an average phosphate reduction of 88%. Moreover, in this case, there was no visible influence of the hydraulic load on the efficiency of phosphate elimination. In contrast to the stable nitrogen elimination during water renewal, phosphate removal was much less stable. In the first half of the study period, the beds provided much higher efficiency, and the phosphate concentrations were much lower than in the second half (Figure 3e).

**Table 3.** Quality of treated wastewater (sample 1) and renewed water (sample 2) in terms of biogenic substances concentration. The letters A, B and C refer to three different hydraulic loads: 70, 100 and 130 L/m$^2$, respectively.

| Value (mg/L) | Sample 1 | | | Sample 2 | | |
| --- | --- | --- | --- | --- | --- | --- |
| | Min | Max | Mean | Min | Max | Mean |
| $NH_4$ (A) | 0.2 | 8.3 | 2.7 | 0.1 | 8.7 | 0.5 |
| $NH_4$ (B) | 0.2 | 42.7 | 5.7 | 0.1 | 11.7 | 1.1 |
| $NH_4$ (C) | 0.3 | 11.4 | 5.5 | 0.1 | 4.4 | 0.6 |
| $NH_4$ mean | 0.2 | 42.7 | 4.7 | 0.1 | 11.7 | 0.8 |
| $NO_3$ (A) | 38.0 | 328.0 | 124.7 | 8.0 | 66.0 | 45.0 |
| $NO_3$ (B) | 17.5 | 450.0 | 141.3 | 3.1 | 357.8 | 51.3 |
| $NO_3$ (C) | 37.4 | 352.0 | 128.4 | 6.0 | 77.2 | 51.5 |
| $NO_3$ mean | 17.5 | 450.0 | 132.1 | 3.1 | 357.8 | 49.4 |
| $NO_2$ (A) | 0.1 | 4.9 | 1.0 | 0.0 | 2.2 | 0.4 |
| $NO_2$ (B) | 0.0 | 3.5 | 0.9 | 0.0 | 2.6 | 0.9 |
| $NO_2$ (C) | 0.0 | 3.5 | 1.1 | 0.1 | 3.1 | 0.6 |
| $NO_2$ mean | 0.0 | 4.9 | 1.0 | 0.0 | 3.1 | 0.7 |
| Total Nitrogen (A) | 7.1 | 88.0 | 41.6 | 1.8 | 17.5 | 12.0 |
| Total Nitrogen (B) | 16.0 | 90.0 | 42.3 | 2.8 | 23.0 | 11.6 |
| Total Nitrogen (C) | 8.4 | 88.0 | 42.7 | 1.7 | 19.0 | 12.3 |
| Total Nitrogen mean | 7.1 | 90.0 | 42.2 | 1.7 | 23.0 | 12.0 |
| $PO_4$ (A) | 17.5 | 34.6 | 27.2 | 0.5 | 9.2 | 3.4 |
| $PO_4$ (B) | 6.5 | 37.3 | 23.4 | 0.6 | 13.3 | 3.0 |
| $PO_4$ (C) | 10.3 | 39.4 | 27.6 | 0.6 | 14.9 | 2.9 |
| $PO_4$ mean | 6.5 | 39.4 | 25.9 | 0.5 | 14.9 | 3.1 |
| Total Phosphorus (A) | 7.1 | 26.0 | 12.7 | - | - | - |
| Total Phosphorus (B) | 3.2 | 26.9 | 12.6 | - | - | - |
| Total Phosphorus (C) | 3.8 | 27.2 | 13.1 | - | - | - |
| Total Phosphorus mean | 3.2 | 27.2 | 12.8 | - | - | - |

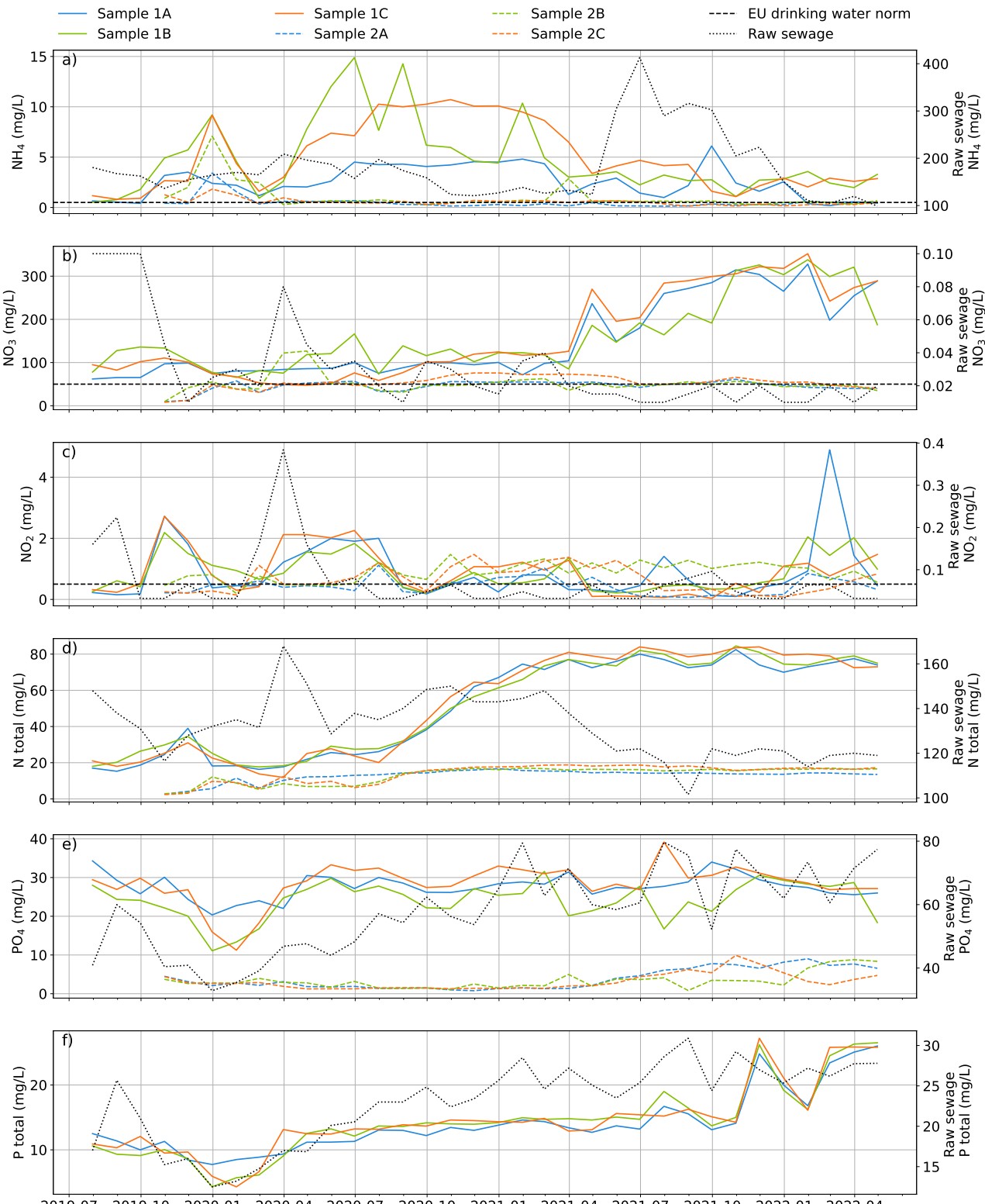

**Figure 3.** Concentration of biogenic substances in the treated sewage (sample 1) and in the renewed water (sample 2). The letters A, B and C refer to three different hydraulic loads: 70, 100 and 130 L/m$^2$, respectively. (**a**) Ammonium (NH$_4$), (**b**) nitrate (NO$_3$), (**c**) nitrite (NO$_2$), (**d**) total nitrogen, (**e**) phosphate (PO$_4$) and (**f**) total phosphorus. The drinking water quality recommendation is taken from the European Council Directive [32].

### 3.3. Transformation of Nitrogen and Phosphorus Compounds in the Renewed Water Reservoir

The reservoir was designed to store the renewed water. However, filling it with fine gravel and cutting it off from the supply of oxygen and light resulted in slight changes in water quality. In order to monitor these changes, the water was periodically analyzed for the content of, inter alia, $NH_4$, $NO_2$, $NO_3$ and $PO_4$. Based on the results from the comparison of the average values of water runoff from the beds and from the buffer reservoir (Table 4), slight changes occurred in each of the monitored components. The $NH_4$ concentration increased from 0.8 to 1 mg/L, $NO_2$ from 0.7 to 0.9 mg/L and $PO_4$ from 3.1 to 5.4 mg/L. Only the concentration of $NO_3$ decreased from the value of 49.4 to 31 mg/L. Only, in this case, there was an improvement in water quality. In the remaining cases, the water quality slightly deteriorated.

**Table 4.** Concentration of biogenic substances in the renewed water reservoir.

| Value (mg/L) | Min | Max | Mean |
|---|---|---|---|
| $NH_4$ | 0.1 | 2.5 | 1.0 |
| $NO_3$ | 0.0 | 67.2 | 31.2 |
| $NO_2$ | 0.1 | 2.8 | 0.9 |
| $PO_4$ | 0.8 | 9.9 | 5.4 |

## 4. Discussion

### 4.1. Removal of Nitrogen and Phosphorus in the Septic Tank

The basic task of the septic tank was to reduce organic pollutants and suspended solids. The HRT of sewage in the tank was 8 days, which means that the sewage was mechanically and partially biologically pretreated in anaerobic conditions. This is confirmed by the reduction in Biological Oxygen Demand and Chemical Oxygen Demand at the level of 70% and of suspended solids at the level of 86%, which was achieved in this tank and presented in the work of Halicki and Halicki [12]. As for TN and TP, the flow of wastewater through the septic tank led to a reduction of around 11%. During conventional biological wastewater treatment, nitrogen and phosphorus are eliminated in sewage treatment plants at the level of up to 30% [30]. This much smaller reduction in the septic tank is mainly due to the fact that the biological treatment in the tank is incomplete (only 70%) and that the decomposition of organic pollutants takes place under anaerobic conditions. A similar reduction in the septic tank is also confirmed by Darby and Leverenz [33], where an average reduction of 12 to 23% for TN and from 0 to 6.7% for TP was found. Considering the fact that the entire pilot plant provided over 90% reduction in TN and TP, the septic tank purification step did not significantly affect the amount of total elimination of these compounds. A septic tank is, however, a necessary element of the cleaning system, mainly due to the removal of suspended solids and organic matter.

### 4.2. Removal of Nitrogen and Phosphorus Compounds in the Process of Wastewater Treatment and Water Renewal

The main goal of the research project, the results of which are presented in this study, was the development of a natural technology for domestic wastewater treatment with water renewal to drinking water quality. As regards nitrogen compounds, European standards limit the concentration of nitrates in drinking water to 50 mg/L and nitrites and ammonium nitrogen to 0.5 mg/L [32]. As shown in Table 3, in the outflow from the pilot plant, the average value for ammonium nitrogen for the entire period was 0.8 mg/L, which slightly exceeds the European standard. However, it should be noted that the pilot plant consisted of three parallel parts (A, B and C) with different hydraulic loads. For the lowest daily load value of 70 L/m$^2$ (part A), the mean value of ammonium nitrogen concentration was 0.5 mg/L, which can generally be considered as meeting the European standard. The same is true of the nitrite concentration: the mean value of all parts is slightly above the drinking water requirements, but the part A effluent concentration meets the requirements. For nitrates, the mean value is within the European standard, and the lowest concentration was

observed in part A (45 mg/L). As can be seen in Figure 3a, ammonium nitrogen fluctuations throughout the year were small and oscillated around the limit value of 0.5 mg/L. This means that in all seasons of the year, the pilot plant with a daily hydraulic load of 70L/m$^2$ was able to guarantee the concentration of ammonium nitrogen in the outflow at the level of drinking water quality. The elimination of nitrates turned out to be slightly less stable. Here, during the entire measuring period, there were cases of exceeding the value of 50 mg/L. However, they occurred mainly in the first research period, and in the last year of the research, the fluctuations were much smaller. The least stable elimination occurred in the case of nitrite concentration, although their mean value for part A was definitely below the required value in drinking water.

Even though the purified water was not intended for drinking water, it was up to standard in terms of nitrogen content. However, a question can be asked whether it is worth striving for such an effective reduction in nitrogen in the renewed water. If we assume that renewed water can be used for: impoundment reuse or environmental reuse, it is of great importance, since nitrogen compounds in the form of nitrite and ammonia (NH$_3$—formed from ammonium nitrogen in high-pH water) are toxic to fish. Moreover, ammonium nitrogen in water causes oxygen reduction [34]. Such a high level of nitrogen elimination is also necessary when discharging the renewed water into groundwater, which is used as a source of drinking water. In such a case, according to American requirements, the renewed water must meet the drinking water requirements [19]. The European standards for the reuse of renewed water, which will enter into force in June 2023, do not provide quality requirements for this use of water—they only provide for its agricultural reuse, in which the removal of nutrients is not limited [6]. The lack of requirements in this regard does not mean that there should be no efforts to introduce other forms of water reuse in the territory of the EU, which, as the study has shown, may meet the criteria of drinking water quality. The lack of requirements in this regard is caused, inter alia, by the lack of appropriate technologies that allow cheap and simple treatment of sewage to the quality of drinking water. For this reason, the presented results should be an incentive for further research in this area, as they show that natural systems are able to obtain the highest-quality cleaning effect.

This is also confirmed by other studies in which natural systems are able to achieve similar effects, e.g., [35,36]. However, not all natural systems achieve such a high efficiency in nutrients removal. Comparing the obtained values in the studied pilot plant with other natural treatment plants, a significant difference can be seen. A similar pilot system for the treatment of domestic wastewater was tested in Mexico [37]. In the outflow from the installation, the average values from the several-month study period were 26 mg/L for ammonium nitrogen and 57 mg/L for total nitrogen. These results differ from the values obtained in the NTSW pilot plant described in this study, where the average value of ammonium nitrogen was 0.8 mg/L and total nitrogen 11 mg/L for the 2.5-year period of the study. Moreover, the use of constructed wetlands for the additional treatment of treated wastewater (tertiary treatment) does not give such a high reduction in nitrogen compounds, as in the case of the described pilot plant [38]. On the other hand, the removal of phosphate from water can be even higher in other natural systems. An example is research on a system combining cascade aeration and treatment in a constructed wetland, in the outflow of which a phosphate concentration of 0.3 mg/L can be obtained [39].

Water renewed in the presented pilot installation was qualitatively similar to drinking water in terms of nutrients content. Moreover, considering its high quality in terms of organic matter content [12], this water can be fully reused in households and for irrigation. The nutrient removal efficiency exceeded the efficiency of natural treatment systems presented in other studies. It can therefore be concluded that the presented installation has fulfilled its task and can be used, especially in places with limited access to water. The main advantages of the described pilot plant are: smaller area of this installation when compared with other plants, as well as very high quality of water in the outflow, which is similar to drinking water. This in turn enables full water reuse. On the other hand, the

construction of such an installation is somewhat complicated. Its operation also requires regular inspection.

## 5. Conclusions

Natural Treatment Systems for Wastewater are, so far, an insufficiently recognized method of effective wastewater treatment. They allow for cheap, ecological and, above all, effective wastewater treatment, as well as for the renewal of water, enabling its reuse. This paper presents over 2.5 years of operation of the NTSW pilot plant, which was built at the Institute of Applied Ecology in Skórzyn (West Poland), and treated wastewater from a building with three permanent residents and up to five office workers. The pilot plant consisted of a wastewater treatment section, namely the septic tank and compost beds, and a water renewal section, namely denitrification beds, phosphorus beds and activated carbon beds. The effectiveness of this installation in removing organic matter from domestic sewage has already been confirmed in the work of Halicki and Halicki [12]. On the other hand, this article presents the reduction in nutrients, namely nitrogen and phosphorus compounds. The purpose of this installation was to treat wastewater and renew water to a level close to that of drinking water. It was assumed that some parameters of treated water, including nitrogen compounds, will achieve the quality of drinking water. In contrast, phosphorus compounds that are not limited in drinking water will be reduced as much as possible during water renewal. With an active area of $3 m^2$ per person and a Hydraulic Residence Time of 6 days (+ 8 days in the septic tank), the pilot plant managed to treat domestic wastewater to a level similar to drinking water. The mean values of nitrogen and phosphorus compounds concentrations for the entire research period were: 0.8 mg/L, 49.4 mg/L, 12 mg/L and 3.1 mg/L for $NH_4$, $NO_3$, TN and $PO_4$, respectively. Thereby, an average reduction of 99.6%, 90.9% and 94.4% for $NH_4$, TN and $PO_4$, respectively, was achieved. Water purified to such an extent can be reused in households (e.g., for flushing toilets or irrigating lawns), which reduces water consumption by up to 50%. This water can also be safely discharged into the environment.

In this study, we show that it is possible to remove biogenic substances from domestic sewage to a level close to that of drinking water. This was achieved in the presented NTSW pilot plant. Similarly, high removal rates were achieved in three different hydraulic loads, namely 70, 100 and 130 $L/m^2$. This installation is cheap to build and to operate. Furthermore, it is characterized by a very low energy consumption. In a world of growing demand for water and increasingly constrained resources, research on such installations should be intensified, and their use should be widely disseminated.

**Author Contributions:** Conceptualization, W.H.; methodology, W.H.; software, M.H.; validation, W.H.; formal analysis, M.H.; investigation, W.H.; resources, W.H.; data curation, M.H.; writing—original draft preparation, W.H. and M.H.; writing—review and editing, W.H. and M.H.; visualization, M.H.; supervision, W.H.; project administration, W.H.; funding acquisition, W.H. All authors have read and agreed to the published version of the manuscript.

**Funding:** This research was funded by the National Center for Research and Development, grant number POIR.01.01.01-00-0805/18.

**Data Availability Statement:** Not applicable.

**Conflicts of Interest:** The authors declare no conflict of interest. The funders had no role in the design of the study; in the collection, analyses, or interpretation of data; in the writing of the manuscript; or in the decision to publish the results.

## Abbreviations

The following abbreviations are used in this manuscript:

| | |
|---|---|
| EU | European Union |
| IES | Instytut Ekologii Stosowanej (Institute of Applied Ecology) |
| HRT | Hydraulic Residence Time |
| NTSW | Natural Treatment System for Wastewater |
| TN | Total Nitrogen |
| TP | Total Phosphorus |

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
