# Peer review of "Effective Removal of Biogenic Substances Using Natural Treatment Systems for Wastewater for Safer Water Reuse"

_water, doi:10.3390/w14233977_

Round 1

Reviewer 1 Report

Very interesting work presented in a clear and systematic way. 

Suggest the following minor modifications.

- Suggest to present the values with one decimal instead of two. One decimal is sufficient for the accuracy of the analytical tools.

- Line 42 requires a reference, after reclaimed water.

- Add the limit of detection of the analytical methods used in the research.

Author Response

1.Request to present the values with one decimal instead of two.We
rounded every value to one decimal.

2.Request to insert a reference at line 42.We did not find a direct citation for
this sentence. However, the next sentence refers to it and we provided a citation
there. Below we inserted another sentence (with a reference), which confrms the
statement in question.
3.Request to add the limit of detection of the analytical methods usedin the research. We provided the measuring range of the analytical methods
in Table 1 (line 160).

Reviewer 2 Report

The findings if this research are of considerable interest and well done. I recommend it to be published after a minor revision.

1. The novelty needs to refinement and should be highlighted in the introduction part.

2. Maybe the author should compare their results clearly with other reported works, highlighting the advantage and disadvantages of their novel composite.

3. The manuscript contains some minor typo/grammar errors, please check all of it.

4. Introduction part, if possible, some important and relative reports references could helped: https://doi.org/10.1016/j.surfin.2022.102006, https://doi.org/10.1016/j.heliyon.2022.e09652.

5. Abstract not targeted; the authors should rephrase it.

6. references should be revised, some of them are incomplete.

Hence, I recommend it accepted for publication after some minor revisions.

Author Response

Request to refine the novelty and to highlight it in the introduction part. We highlighted the novelty of the presented pilot plant in the last paragraph of the introduction.

Request to compare our results clearly with other reported works and to highlight the advantages and disadvantages of our approach. A comparison with similliar studies is presented in the penultimate paragraph in the Discussion. We also added there a short highlight of advantages and disatvantages of our approach in the last paragraph of this section.

Request to check the manuscript in terms of typo-grammar errors. The manuscript was again proof-read by the authors. Further, we additionally consulted the whole document with an english native speaker. Therefore some aditional small changes occur in the manuscript.

Request to supplement the introduction with two reports. We thank the Reviewer for the papers attached, but we did not find a suitable place in the manuscript to cite them.

Request to rephrase the abstract. We made several small changes in the abstract. We hope that it is now better targeted. If more changes are required we would appreciate a more specific remark.

Request to revise the references. We revised the references and completed them where sufficient information could be found.

Reviewer 3 Report

In this article, the authors presented research on removal of biogenic substances using natural treatment systems for wastewater for a safer water reuse. Although the similar topic has already been discussed by the authors, the data obtained is interesting.

The manuscript requires a correction. Detailed comments:

1.       Line: 10-12 (abstract) - Explanations of all abbreviations in the abstract should be added.

2.       The authors could better emphasize the novelty of the research carried out.

3.       Materials and Methods - Please add the geographical coordinates of the place where the research was conducted.

4.       Consider adding a map with the location of the points where the research was carried out.

5.       Please add information when and how many times samples were taken for testing (how many series were made?)

6.       It is important to check that the writing text clearly expresses and explains each idea and result obtained.

7.       Conclusions- The conclusions needs improvement - highlight the most important findings and identify the added value of the main finding.

8.       There are no references to the Water (only item 10) to which the authors have submitted their manuscript. Add articles from this journal to emphasize that the submitted manuscript is a good fit.

Author Response

Request to add explanations to all abbreviations in abstract. We thank the Reviewer for this remark. We explained all abbreviations. 

Request to emphasize the novelty of the research carried out. We emphasied the novelty of the presented pilot plant at the end of the Introduction.

Request to add the geographical coordinates of the place where the research was conducted. Geographic coordinates were provided in line 108.

Request to add a map with the location of the points where the research was carried out. In this study we present removal efficiency of one pilot plant. Instead of providing an entire map with only one point, we mention the link to the website (line 108), where an interactive map with the location of the studied plant can be found. 

Request to provide information when and how many times samples were taken for testing. Information on the frequency of sampling can be found in the ``Sampling and quality assessment'' subsection. Following the request of the Reviewer, we added to this subsection the time span of this study.

Request to check that the writing text clearly expresses and explains each idea and result obtained. The manuscript was again proof-read by the authors. Further, we additionally consulted the whole document with an english native speaker. Therefore some aditional small changes occur in the manuscript.

Request to improve the conclusions - highlight the most important findings and identify the added value of the main finding. We followed the Reviewer's instructions and provided a highlight of the most important findings at the end of the section.

Request to add more articles from the Water journal. We included one more article, which shows a recent developement of a NTSW pilot plant on the Canary Islands.

Round 2

Reviewer 3 Report

The comments are addressed properly and necessary corrections have been done. The manuscript can be accepted.